# LAYERCOMPOSER: INTERACTIVE PERSONALIZED T2I VIA SPATIALLY-AWARE LAYERED CANVAS

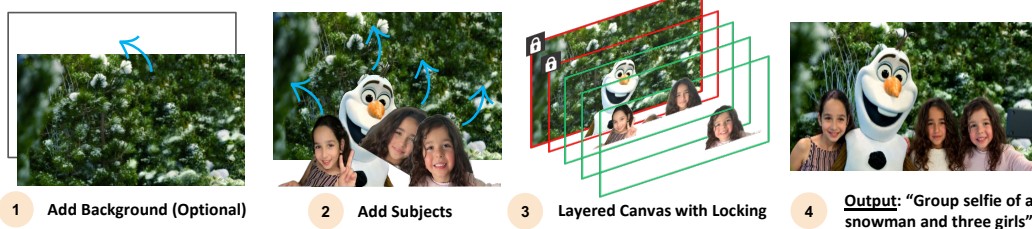

Figure 1: **LayerComposer** introduces an **interactive personalization** paradigm that enables a Photoshop-like experience for multi-subject T2I generation. It allows users to *place*, *resize*, and *lock* subjects on the proposed **layered canvas**. A new *locking* function is provided such that locked subjects (*e.g.*, background, snowman) are preserved with only necessary lighting adjustments, while unlocked subjects are flexibly injected into the scene with variations guided by the text prompt.

## ABSTRACT

Despite their impressive visual fidelity, existing personalized generative models lack interactive control over spatial composition and scale poorly to multiple subjects. To address these limitations, we present *LayerComposer*, an interactive framework for personalized, multi-subject text-to-image generation. Our approach introduces two main contributions: (1) a *layered canvas*, a novel representation in which each subject is placed on a distinct layer, enabling occlusion-free composition; and (2) a *locking mechanism* that preserves selected layers with high fidelity while allowing the remaining layers to adapt flexibly to the surrounding context. Similar to professional image-editing software, the layered canvas allows users to *place*, *resize*, or *lock* input subjects through intuitive layer manipulation. Our versatile locking mechanism requires no architectural changes, relying instead on inherent positional embeddings combined with a complementary data sampling strategy. Extensive experiments demonstrate that *LayerComposer* achieves superior spatial control and identity preservation compared to the state-of-the-art methods in human-centric personalized image generation.

## 1 INTRODUCTION

The advent of large-scale text-to-image (T2I) diffusion models (Rombach et al., 2022) has marked a pivotal moment in digital content creation, enabling the synthesis of complex, high-fidelity images from simple textual descriptions. This breakthrough has spurred a wave of research into personalization, which aims to create content containing specified identities in unseen contexts. Textual Inversion (Gal et al., 2023), DreamBooth (Ruiz et al., 2023), and IP-Adapter (Ye et al., 2023) have made significant strides in this domain in recent years.

Despite their progress, the creative potential of existing personalized generative models is largely hindered due to two critical shortcomings: a lack of interactive spatial control and a fundamental inability to scale efficiently to multiple identities. First, to enable spatial guidance, current approaches rely on frameworks like ControlNet (Zhang et al., 2023). These approaches require users to generate auxiliary control maps like pose skeletons or depth maps, which unfortunately fragments the creative process. Second, to achieve multi-identity personalization, existing techniques (Ye et al., 2023; Chen et al., 2025; Qian et al., 2025b) encode identity images into fixed-length token sequences that are then

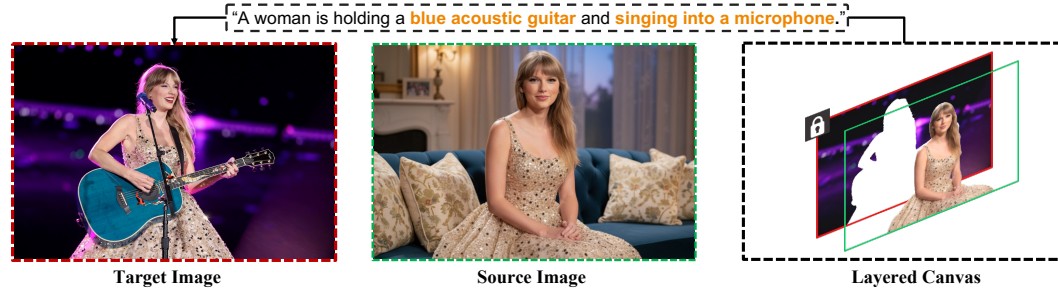

Figure 2: **Locking-Aware Data Sampling Strategy.** During training, layers in **layered canvas** (the input to our LayerComposer) are extracted from multiple images within the same identity. Locked layers (e.g., background) are sampled directly from the target image, resulting in pixel alignment in input-target pair and thus preserving fidelity. In contrast, unlocked layers (e.g., the woman) are sampled from other source images, enabling variation guided by the text prompt while maintaining identity. Data augmentations are applied to all layers during training, ensuring that both locked and unlocked layers can adapt to context at inference (e.g., lighting).

concatenated into longer conditioning embeddings. Such a method is constrained by a memory cost that increases linearly with respect to the number of personalized subjects, making the composition of many unique subjects expensive and even infeasible with a large number of identities. Together, these limitations highlight the urgent need for a new paradigm with better interactivity and scalability.

To meet this demand, we introduce an **interactive personalization** paradigm. Inspired by professional editing software (*e.g.*, Photoshop), this paradigm positions the user as an art director, who can intuitively compose a desired scene by *placing* and *resizing* multiple distinct input subjects on a canvas, as shown in Fig. 1(1-2). This canvas-based approach is designed for better interactivity and scalability, allowing the composition of many identities within a scene. The resulting canvas configuration acts as a "visual template", defining an optional background, multiple identities, and their spatial arrangement.

To bring this template to life, we introduce *LayerComposer*, a generative model specifically designed to render a canvas into a single, coherent, and high-fidelity image as in Fig. 1(4) in a feedforward manner. The core of LayerComposer is a **layered canvas**, a novel input representation composed of a few spatially aware RGBA layers where each layer defines a personalized subject as illustrated in Fig. 1(3). While its composition can be conceptualized as a collage (Sarukkai et al., 2024), layered canvas offers three critical advantages. First, by preserving each subject on a separate layer, it resolves occlusions that occur in the traditional collages where subjects overlap—an issue that frequently occurs in multi-subject personalization. Second, our diffusion model uses the token sequence from the layered canvas for conditioning. This token sequence is constructed via the **transparent latent pruning** strategy, where we extract and concatenate only the valid tokens corresponding to non-transparent (non-zero alpha) regions from all layers. This strategy decouples the length of the conditioning sequence from the number of identities, enabling more efficient composition with a scalable number of personalized elements.

Last but not least, our layered canvas provides the foundation for a novel **locking mechanism**, which offers fine-grained control over content preservation. Users can selectively *lock* any layer, constraining the model to preserve its visual content with high fidelity, while permitting necessary lighting adjustment adjustments. In parallel, the unlocked layers remain free to vary, allowing their appearance or pose to be synthesized according to the context. This capability is critical for practical personalized T2I, such as preserving a character's pose or maintaining a specific background while regenerating the rest of the scene as shown in Fig. 1(3-4). To achieve the locking mechanism, LayerComposer employs a **model-data co-design**, where we leverage the inherent positional embeddings of pretrained models guided by a complementary data sampling strategy (Fig. 2), requiring no architectural modifications.

LayerComposer therefore provides a scalable and interactive solution in which users can compose many identities with high-fidelity control simply by arranging and locking layers on a canvas. Our **contributions** are as follows:

- We propose an interactive personalization paradigm for T2I generation, empowering users to act as active directors by directly placing, resizing, and locking subjects on a canvas.
- We introduce layered canvas, a novel layered input representation that addresses the scalability bottleneck through transparent latent pruning, and handles occlusion issues by its layered design.
- We present a novel locking mechanism achieved by a simple yet effective model-data co-design strategy that does not require architectural changes.
- Through comprehensive evaluations, we demonstrate that LayerComposer achieves state-of-the-art compositional control and fidelity compared to state-of-the-art personalization methods.

## 2 RELATED WORK

**Personalized Generation.** Personalization methods have shifted from expensive per-concept tuning (Gal et al., 2023; Nitzan et al., 2022; Ruiz et al., 2023) to recent adapter-based solutions (Ye et al., 2023; Li et al., 2024; Wang et al., 2024b; Gal et al., 2024; Guo et al., 2024; Qian et al., 2025a; Patashnik et al., 2025), which enable efficient personalization while keeping the base diffusion model frozen. Despite their efficiency, these methods provide limited interactive spatial control and suffer from a fundamental scalability bottleneck when composing multiple subjects.

**Spatial Control in Generation.** A broad range of conditioning mechanisms has been proposed to improve controllability. Pose-guided methods such as ControlNet (Zhang et al., 2023) and T2I-Adapter (Mou et al., 2024) inject structural cues, while region-based approaches provide layout guidance through bounding boxes (Li et al., 2023; Zheng et al., 2023; Dahary et al., 2024; Song et al., 2023; Lee et al., 2024; Zhang et al., 2025) or segmentation masks (Yang et al., 2023; Liu et al., 2025; Chen et al., 2024). Although these approaches excel at either identity preservation or layout specification, they typically struggle to achieve both simultaneously. Collage-based techniques such as CollageDiffusion (Sarukkai et al., 2024), NoiseCollage (Shirakawa & Uchida, 2024), and HiCo (Cheng et al., 2024) demonstrate spatial control but often introduce artifacts, are limited to occlusions, and some are computationally expensive (e.g., requiring $O(N)$ passes per step).

**Multi-Concept Personalization.** Generating images that faithfully integrate multiple personalized concepts remains challenging. Optimization-based approaches disentangle concepts (Kumari et al., 2023; Avrahami et al., 2023; Garibi et al., 2025) or train multiple LoRAs (Po et al., 2024; Kong et al., 2024). Optimization-free approaches rely on lightweight adapters (Xiao et al., 2025a; Wang et al., 2024a; Han et al., 2024; Dalva et al., 2025; Chen et al., 2025; Qian et al., 2025b), but suffer from linear complexity growth as subjects increase. General in-context image generation (Xiao et al., 2025b; Wu et al., 2025c; Comanici et al., 2025; Wu et al., 2025a) supports arbitrary concepts but offers limited interactivity, no selective preservation, and limited human generation quality. LayerComposer advances the human-centric personalization literature by a layered canvas that achieves scalable personalization, resolves occlusion ambiguity, and supports a locking mechanism for selective, high-fidelity preservation.

## 3 LAYERCOMPOSER

### 3.1 LAYERED CANVAS

LayerComposer is a controllable text-to-image generation framework that offers an interactive personalization experience, enabling users to control both the spatial composition and the appearance of multiple subjects (identities and optional background). Concretely, the framework conditions a pretrained diffusion model on two inputs: (1) a text prompt that specifies global image content and high-level semantics, and (2) a layered canvas that jointly encodes spatial and visual guidance of the subjects, augmented with a binary locking flag that determines the degree of fidelity preservation. This design ensures that LayerComposer adheres faithfully to the user's compositional intent: preserving locked subjects with maximum fidelity while harmonizing the overall output into a globally coherent scene.

The **layered canvas** is represented by a set of RGBA layers $L = \{l_1, \cdots, l_N\}$ and a corresponding set of binary locking flags $B = \{b_1, ..., b_N\}$, where $N$ denotes the number of layers. Each RGBA layer $l_i$ encodes the information of one subject. The RGB channels provide visual reference of the subject while the alpha channel defines its spatial mask, indicating the valid regions of presence.

Figure 3: **LayerComposer Pipeline.** LayerComposer conditions a diffusion model on both a text prompt and a **layered canvas**. The canvas consists of multiple layers that can be either locked or unlocked. Each layer is first encoded using the VAE. Next, the positional embeddings are added according to the layer's locking status: locked layers share the same positional embeddings as the noisy latent $[0, x, y]$, while each unlocked layer is assigned a unique layer index $j$ in its positional embeddings $[j, x, y]$. $j$ distinguish unlocked layers when they overlap. Finally, a **transparent latent pruning** is performed to retain only the latents in non-transparent regions per layer, while discarding the others (gray boxes) for scalable personalized generation.

Subsequently, this mask is used to identify the valid tokens that will be used for generation, as detailed in Sec. 3.2.

The **locking flag** $b_i$ determines whether the layer should be strictly preserved or allowed to adapt. When a layer is locked ($b_i = 1$), the model is constrained to render the subject in the layer with maximum fidelity, permitting only minimal variations (*e.g.*, lighting or shading adjustments) to ensure seamless integration with the rest of the scene. In contrast, when a layer is unlocked ($b_i = 0$), the subject may be flexibly adapted to the surrounding context while still retaining its semantic identity. This mechanism balances fidelity and adaptability, ensuring that user-specified subjects can be either preserved exactly or reinterpreted creatively depending on the desired outcome.

## 3.2 LAYERCOMPOSER PIPELINE

LayerComposer builds on a pretrained latent-based diffusion transformer (DiT) (Peebles & Xie, 2023), as illustrated in Fig. 3. Our framework first encodes the input layered canvas into conditional latent tokens, which are then concatenated with noisy latent tokens to achieve personalization. To improve scalability to multiple subjects, we introduce a **transparent latent pruning** strategy, which discards the tokens corresponding to transparent (zero alpha value) regions and retains only those from valid spatial locations across all layers. To enable the locking functionality, we assign distinct positional embeddings to the latents from each layer, encoding both their specific location and locked status. The full pipeline is described below.

**Layer Latent Extraction.** For each input layer $l_i \in L$, we first encode the RGB content using the pretrained VAE encoder to obtain layer latents $z_i = \mathcal{E}(l_i^{\text{RGB}}) \in \mathbb{R}^{H' \times W' \times D}$ where $\mathcal{E}$ is the VAE encoder, $H'$ and $W'$ are the spatial dimensions in latent space, and $D$ is the feature dimension.

**Positional Embedding with Locking.** To enable the locking mechanism, we introduce a simple yet effective positional embedding scheme: each layer latent $z_i$ is augmented with a 3D positional embedding that encodes both its spatial location and locking status:

$$\text{pos}_i = \begin{cases} [0, x, y] \in \mathbb{R}^3, & b_i = 1 \quad \text{(locked)} \\ [j, x, y] \in \mathbb{R}^3, & b_i = 0 \quad \text{(unlocked)} \end{cases} \tag{1}$$

where $(x, y)$ are the spatial coordinates in the latent space and $b_i \in \{0, 1\}$ is a binary locking flag. For the locked layers ($b_i = 1$), we fix the layer index (*i.e.*, the first positional dimension) to 0. As a result, all locked subjects share the same layer as the noisy latent tokens, which also use the positional embeddings $[0, x, y]$. The motivation is that the pretrained diffusion models exhibit strong spatial and visual consistency when conditioned on nearly clean latent tokens (Ho et al., 2020). Leveraging this property, we reuse the same positional embeddings for locked subjects, which empirically yields

highly faithful preservation across denoising steps. In contrast, for the unlocked layers ($b_i = 0$), each of them is assigned a unique index $j$ in the first dimension to separate each unlocked subject in a distinct layer, where $j \in \{1, 2, \ldots, |\{ i \mid b_i = 0 \}|\}$. This separation is used to avoid mixed appearance when two subjects overlap in the canvas.

**Transparent Latent Pruning.** To increase the scalability of multi-subject personalization, we introduce a transparent latent pruning strategy that selectively retains the latent tokens from valid spatial locations according to the alpha channel, while discarding the rest. Concretely, for each layer's alpha channel $l_i^\alpha$, we first downsample it to the latent resolution using nearest-neighbor interpolation:

$$\alpha_i^{\text{latent}} = \text{NearestResize}(l_i^\alpha) \in \mathbb{R}^{H' \times W'}. \tag{2}$$

We then apply alpha-based masking to the latent tokens $z_i$, keeping only those in regions with non-zero alpha values:

$$z_i^{\text{valid}} = \text{Concat}(\{z_i(x, y) | \alpha_i^{\text{latent}}(x, y) = 1\}), \tag{3}$$

where $z_i(x, y)$ and $\alpha_i^{\text{latent}}(x, y)$ denote the latent token and its corresponding alpha value at spatial coordinate $(x, y)$.

Prior methods (Chen et al., 2025; Qian et al., 2025b) concatenate token sequences of fixed length from all input images, leading to a memory cost that increases linearly with respect to the number of personalized subjects $N$. In contrast, our transparent latent pruning strategy makes the length of the token sequence proportional only to the non-transparent content area, yielding substantial efficiency improvements when handling many personalized subjects.

**Layer Conditioning Integration.** Finally, we construct the conditional latents of the layer canvas by aggregating the pruned latents from all layers: $z_{\text{cond}} = \text{Concat}(z_1^{\text{valid}}, z_2^{\text{valid}}, \ldots, z_N^{\text{valid}})$, which is then concatenated with the noisy image latents $z_t$ to form the latent input of the DiT model.

## 3.3 LAYERCOMPOSER TRAINING

As aforementioned, our model treats locked layers and unlocked layers differently. In this section, we detail how we sample both types of layers, and how is the model trained.

**Locking-Aware Data Sampling Strategy.** LayerComposer training employs a locking-aware data sampling strategy, illustrated in Fig. 2. This training requires a multi-image-per-scene dataset described in Sec. 4.1. Each scene consists of an image set containing the same identities. For each training sample, one ground-truth image is randomly selected as the target, $I^{\text{target}}$, while the remaining images in the scene serve as sources. Each image is segmented into subjects (e.g., humans, backgrounds), with each subject $i$ assigned to its own layer $l_i$.

The input layered canvas is constructed as follows. We initialize $L = \{\}$. A random subset of layers from the target image is added to $L$ and marked as locked ($b_i = 1$), yielding $L = \{l_i^{\text{target}} \mid b_i = 1\}$. The remaining layers for subjects that are not selected are marked as unlocked ($b_i = 0$) and are sampled from the corresponding layers from source images within the same scene. Unlike locked layers, which directly copy content from $I^{\text{target}}$, unlocked layers provide cross-image appearance references without pixel-level correspondence.

In summary, locking-aware data sampling assigns locked layers directly from the target image and unlocked layers from other images in the same scene. This design compels the model to preserve the fidelity of locked content to the maximum extent, while allowing variation in the unlocked layers.

**Layer-Conditioned Finetuning.** We adapt the pretrained model by finetuning it with LoRA (Hu et al., 2022). Specifically, we train the LoRA adapters $\theta$ on the attention layers of the DiT backbone. The parameters are optimized using a flow matching loss (Lipman et al., 2023):

$$\mathcal{L}_{\text{cond}} = \mathbb{E}_{t \sim (0,1), z_0, z_1, z_{\text{cond}}, P} \left[ \left\| v_\theta(z_t, t, z_{\text{cond}}, P) - (z_1 - z_0) \right\|^2 \right], \tag{4}$$

where $v_\theta(\cdot)$ is the predicted velocity, $z_1$ and $z_0$ are the latents of the target image and the sampled noise, $z_t$ is the noisy latents at timestep $t$ of the target image $I^{\text{target}}$, $z_{\text{cond}}$ is the conditional latents of our layered canvas $L$, and $P$ is the text prompt, respectively.

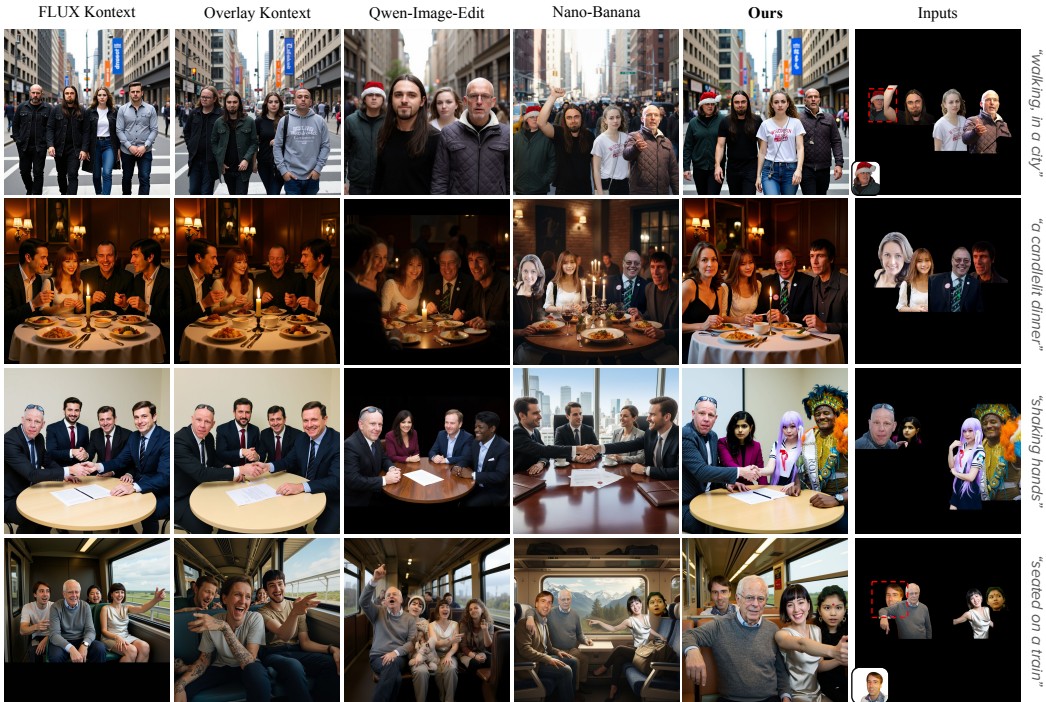

Figure 4: **Qualitative Comparison in Four-person (4P) Personalization.** While state-of-the-art baselines frequently distort, omit subjects, or produce unnatural copy-pasted artifacts, LayerComposer consistently generates high-fidelity and coherent compositions, faithfully preserving teir identities and spatial arrangement of all subjects. Crucially, our approach excels even when subjects are partially occluded in the input (shown in red boxes in 1st and 4th rows) because of our unique layered canvas.

## 4 EXPERIMENTS

### 4.1 EXPERIMENTAL SETUP

**Training Dataset Curation.** Our training set comprises ∼32M in-house images across 6M scenes, focusing on human subjects. We filter the data to ensure each scene contains at most 4 identities, to exclude low-resolution, low-quality faces. To construct the layered training data, we apply internal instance human segmentation to extract each human as a distinct layer and leave the rest as background. When constructing the input layered canvas in each training step, we apply data augmentations to each layer, including random scaling, shifting, and color perturbations.

**Training Details.** We train a LoRA with a rank of 512 on the frozen FLUX Kontext (Labs et al., 2025) using the AdamW optimizer (Loshchilov & Hutter, 2019). The model is trained for 200K iterations with a constant learning rate of $1{\times}10^{-4}$, a batch size of 32, and at a $512{\times}512$ resolution. The entire training took 4 GPU days on 4 nodes, each with 8 A100 GPUs.

**Evaluation Details.** We evaluate at a $1024{\times}1024$ resolution using 128 images from FFHQ-in-the-wild (Karras et al., 2019) as identity inputs. FFHQ is a public, single-frame dataset and is not included in our training. There are 32 prompts for each benchmark. All evaluations are conducted with 28 denoising steps for our model, without any per-prompt tuning or post processing. Quantitatively, following the previous arts, we adopt a set of widely used metrics in personalized T2I. Identity preservation is evaluated with ArcFace (Deng et al., 2022) through the Insightface library (Contributors, 2024), text alignment with prompts is assessed by VQAScore (Lin et al., 2024), and image quality is measured by HPSv3 (Ma et al., 2025). A user study is also performed to pick the best generation among all methods per prompt that reaches the balance among identity preservation, prompt following, and image quality. Check *Appendix* for the evaluation details of all baselines.

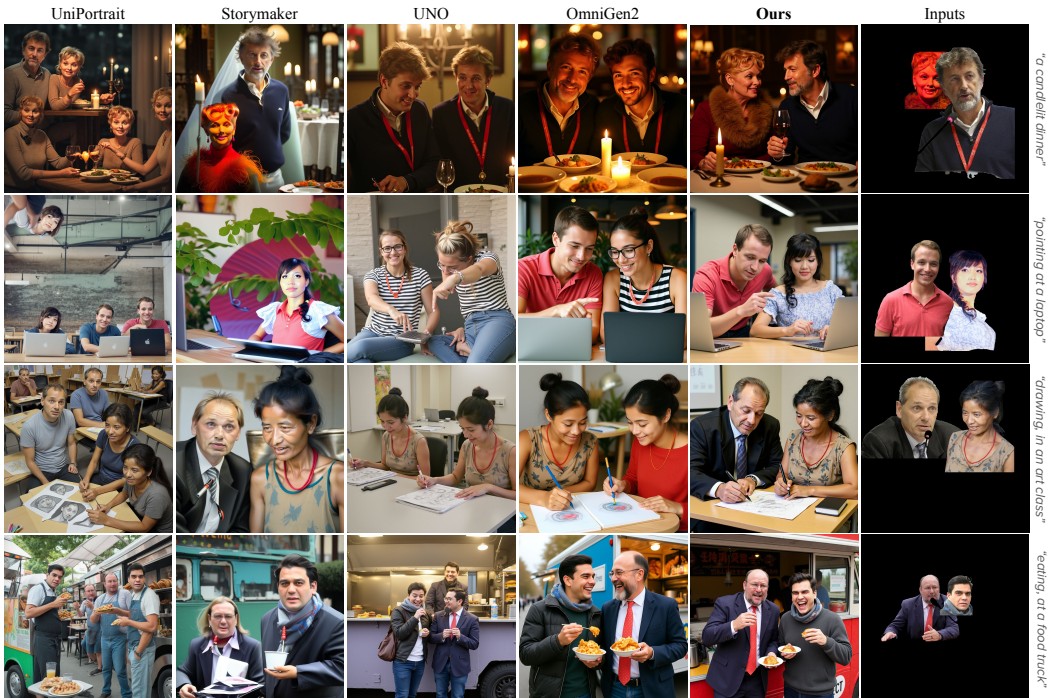

Figure 5: **Qualitative Comparison in Two-Person (2P) Personalization.** When personalizing an image with two subjects, competing methods often fail to compose a coherent, interactive scene. This frequently results in missing, duplicated, or distorted subjects and unrealistic interactions. In contrast, LayerComposer produces visually coherent, high-fidelity scenes where both subjects are present and naturally interacting with each other and their surroundings, while preserving their distinct identities.

## 4.2 BASELINE COMPARISONS

**Four-Person (4P) Personalization.** Most existing personalization methods struggle to scale beyond two persons due to the linear growth in computation and memory with the number of subjects. This bottleneck limits their applicability to challenging but relevant real-world use cases, such as 4P personalization. In contrast, LayerComposer, enabled by our novel layered canvas, natively supports multi-subject personalization beyond the restrictive two-person setting without any prohibitive overhead.

We benchmark LayerComposer in the 4P setting against FLUX Kontext (Labs et al., 2025), Overlay Kontext (a.k.a, Place it) (ilkerzgi & gokaygokay, 2025), Qwen-Image-Edit (Wu et al., 2025a), and Gemini 2.5 Flash Image (a.k.a Nano-Banana) (Comanici et al., 2025). LayerComposer shows significantly stronger performance in human-centric personalization tasks. As shown in Fig. 4, LayerComposer generates high-quality images that faithfully follow user-specified spatial layouts while effectively preserving the identities of the input subjects. More crucially, in multi-subject personalization, where the number of personalized subjects increases, occlusion naturally arises. LayerComposer also excels in the presence of occlusion due to our layered canvas strategy. Baseline approaches, however, often fail under such conditions.

Quantitatively, as reported in Tab. 1, LayerComposer achieves the highest identity preservation as indicated by ArcFace and the highest image quality assessed by HPSv3, maintaining a strong level of prompt following gauged by VQAScore. LayerComposer was also liked in most cases in the user study (48.96% v.s. 34.46% for Nano-Banana), and significantly outperforms other strong baselines.

**Two-Person (2P) Personalization.** Unlike a few canvas-based approaches that can handle up to four persons as discussed above, most existing personalization methods are designed specifically for 2P personalization. State-of-the-art 2P methods, including UniPortrait (He et al., 2025), Story-Maker (Zhou et al., 2024), UNO (Wu et al., 2025c), and OmniGen2 (Wu et al., 2025b), are evaluated

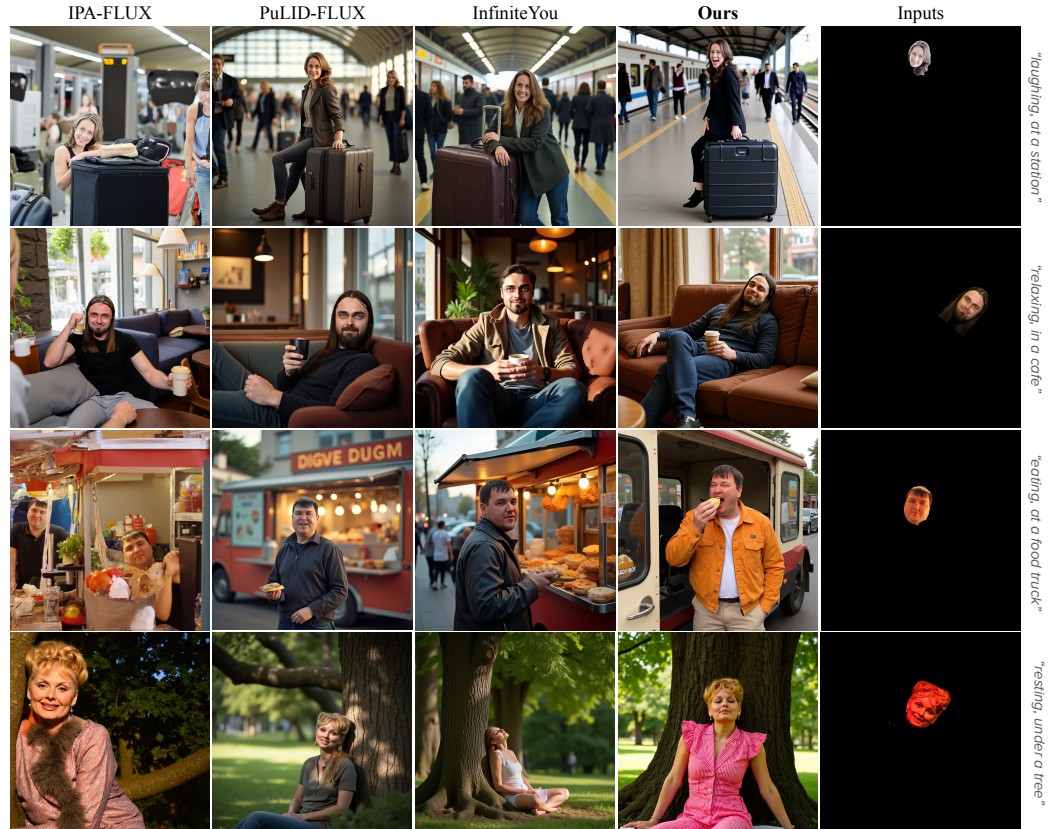

Figure 6: **Qualitative Comparison in Single-Person (1P) Personalization.** State-of-the-art 1P personalization approaches tend to inject the reference face identity with limited flexibility, resulting in copy-pasted effects. In contrast, LayerComposer generates realistic outputs, faithful to both the human identity and text prompt. Notably, our method captures diverse expressions (e.g., *laughing*, 1st row), handles challenging states such as *relaxing* (2nd row), and supports diverse activities like *eating* (3rd row) and *closed eyes* (4th row), which require complex body poses or expressive facial gestures.

in Fig. 5. As observed, prior methods often fail to produce coherent scenes with both subjects correctly placed and interacting naturally: some omit one subject, others duplicate it, and many yield identity not preserved. In contrast, our approach generates high-fidelity, prompt-aligned scenes where both identities are faithfully preserved and their interactions look more natural. As demonstrated in Tab. 1, LayerComposer is significantly preferred by users, achieves the highest identity preservation, and delivers image quality and text adherence on par with OmniGen2, findings that are consistent with our qualitative comparisons.

**Single-Person (1P) Personalization.** Prior to multi-subject personalization, single-person generation was the focus in personalization. To evaluate LayerComposer on 1P benchmark, we compare it to several leading methods developed built on top of FLUX.1 dev (Black Forest Labs, 2024), including IP-Adapter (Ye et al., 2023), PuLID (Guo et al., 2024), and InfiniteYou (Jiang et al., 2025). As illustrated in Fig. 6, competing methods often directly inject the input identity with limited pose and expression variations and are often unable to follow diverse text prompts. In contrast, LayerComposer produces coherent natural generations following prompts with diverse facial expressions. Quantitatively, Tab. 1 verifies that ours excels at prompt following and is preferred by a significantly larger portion in the user study. See §D for additional visual examples.

## 4.3 ABLATION AND ANALYSIS

Figure 7: **Ablation study.** Locking can be applied to any subject, preserving the selected subjects with only necessary lighting and shading adjustments (e.g., reducing head reflection in the right man) in the final output. The layered canvas resolves occlusion issues, without which the overlapped details might be lost (e.g., the hat of the left woman).

Since our contributions primarily lie on the control side—specifically the locking mechanism and the layered canvas—it is most intuitive to evaluate them qualitatively, as quantitative differences are less pronounced. Their effectiveness in preservation and occlusion handling is shown in Fig. 7.

**Effect of Locking Mechanism.** To demonstrate the effect, we progressively lock each input layer. A locked layer preserves the pose of the subject—while the model applies only outpainting and subtle lighting changes. We highlight that this is different from the masked inference, where the masked regions will not be updated at all. In terms of our unlocked layers, they will be flexibly adjusted based on the locked ones and the broader context. See another example in Fig. II for the effect of locking.

**Effect of Layered Canvas.** Without the layered canvas, the model is trained on a single collage image as the conditioning input, shown as "Inputs" in Fig. 7. As seen in the "w/o layered canvas" column, *e.g.*, occlusion in the collage causes missing information. For

| Method | ArcFace ↑ | VQAScore ↑ | HPSv3 ↑ | User Rate(%) ↑ |
|---|---|---|---|---|
| Four-Person (4P) Personalization | | | | |
| FLUX Kontext | 0.217 | **0.869** | 12.3 | 6.25 |
| Overlay Kontext | 0.251 | 0.828 | 11.2 | 2.08 |
| Qwen-Image-Edit | 0.236 | **0.869** | 11.6 | 1.04 |
| Nano-Banana | 0.434 | 0.826 | 10.4 | 36.46 |
| **Ours** | **0.533** | 0.840 | **12.5** | **48.96** |
| Two-Person (2P) Personalization | | | | |
| UniPortrait | 0.536 | 0.723 | 8.01 | 0 |
| StoryMaker | 0.542 | 0.523 | 4.97 | 0 |
| UNO | 0.071 | **0.870** | 11.2 | 0 |
| OmniGen2 | 0.121 | 0.828 | **12.8** | 16.67 |
| **Ours** | **0.547** | 0.796 | 11.6 | **83.33** |
| Single-Person (1P) Personalization | | | | |
| IPA-FLUX | 0.453 | 0.790 | 9.88 | 9.38 |
| PuLID-FLUX | **0.639** | 0.859 | 11.5 | 9.38 |
| InfiniteYou | 0.528 | 0.853 | **13.2** | 15.63 |
| **Ours** | 0.487 | **0.893** | 12.5 | **65.63** |

Table 1: **Quantitative comparison across different personalization benchmarks.** LayerComposer ranks among the top two methods in image quality across benchmarks according to HPSv3. On multi-subject benchmarks, it substantially outperforms other leading approaches in identity preservation as measured by ArcFace. Notably, because ArcFace tends to reward faces with the same expression and head pose, our score on 1P generation is lower than baselines that tend to copy paste the input face. Our strong VQA score in 1P demonstrates superior adherence to text prompts compared to competing baselines. User studies in rating the overall best method for each prompt show that our method is favored across all benchmarks.

example, the ball on the Christmas hat disappears from the left woman. By contrast, our layered canvas explicitly handles occlusion and prevents such artifacts. We also show that the layered canvas is versatile and can accept an optional background as an additional input layer in §D.1.

## 5 CONCLUSION

In this paper, we introduced LayerComposer, a novel and effective framework for interactive personalized text-to-image generation. By treating user inputs as a set of spatially-aware layers, our method provides direct occlusion-free control over the composition of multiple personalized subjects. The proposed locking mechanism further refines this control, enabling high-fidelity subject preservation in the locked layers, while allowing creative variance in the unlocked layers. Our experiments demonstrate that LayerComposer surpasses existing methods in both spatial control and identity preservation, offering a more intuitive and powerful tool for creative expression. We believe that LayerComposer, specifically the layered canvas paradigm, opens the door to many exciting and meaningful future work. See *Appendix* for discussions on limitations of this work.

ETHICS STATEMENT

First, regarding the images used for training, we relied exclusively on our licensed datasets with heavy NSFW filtering. Second, for benchmarking, we used the publicly available dataset FFHQ-in-the-Wild—collected in StyleGAN (Karras et al., 2019)—instead of unlicensed photographs. Finally, we acknowledge that our model can generate synthetic images that may be misused with harmful intent. In accordance with our internal policy, we will not open-source the model at this time. For API access, we will implement input safeguards and prompt filtering mechanisms to mitigate potential misuse.

REPRODUCIBILITY STATEMENT

We describe our dataset curation process and provide sufficient details of training and evaluation in Sec. 4.1 to ensure reproducibility. Our model is built on top of FLUX.1 Kontext (Labs et al., 2025), which is publicly available.

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

# LayerComposer: Interactive Personalized T2I via Spatially-Aware Layered Canvas

## — Supplementary Material —

## A  BENCHMARK DETAILS

### A.1  BASELINES

**4P Personalization.** For all baselines, we use the collaged image as input, as they do not support a layered canvas with spatial control like ours. We use the same prompt for FLUX Kontext (Labs et al., 2025), Qwen-Image-Edit (Wu et al., 2025a), Nano-Banana (Comanici et al., 2025), and our Layer-Composer. For Overlay Kontext (ilkerzgi & gokaygokay, 2025), we follow the official instructions to add the "place it" trigger phrase before the generation prompt: `"Place it.{prompt}"`.

**2P Personalization.** We use the same prompt for all baselines except OmniGen2, which requires in-context instruction: `"The first person is image 1 and the second person is image 2.{prompt}."`

**1P Personalization.** All baselines use the cropped head as input. For our method, we place the same cropped head into a black canvas to be compatible with our training.

### A.2  AUTOMATED BENCHMARKING PIPELINE

Layered canvas is primarily designed for interactive personalization. However, for benchmarking purposes, we built an automated canvas creation pipeline so that evaluation can be performed without any human intervention. The pipeline works as follows: for each prompt, we first run the FLUX.1 dev model to generate a prior image. We then apply face detection, obtaining relevant face bounding boxes. Next, we detect the bounding boxes of all input subjects. Finally, each input face is resized and placed according to the size and location of each prior face, yielding a canvas that serves as the input for the evaluation of our model. The collage image is the composition generated from the layered canvas, which is then used for the collage-based baselines. This automated pipeline is used across all benchmarks in this paper.

## B  LIMITATION AND FUTURE WORK

LayerComposer, despite its innovative personalization paradigm, suffers from limitations originating from data quality and the diffusion backbone.

**Reasoning limitation..** The method sometimes struggles with complex reasoning, particularly when the generated image requires a sophisticated spatial relationship between the humans and the background. For example, LayerComposer fails to correctly place foreground humans in the chairs in a given background. As future work, we argue that this limitation can be addressed by integrating the strong reasoning capabilities of Vision Language Models (Bai et al., 2023; 2025) into the personalized generation process.

**Beyond 4-Person (>4P) Generation Limitations.** LayerComposer in principle supports any number of subjects in the layered canvas. However, its performance is currently limited in scenarios with more than four people, due to two factors. First, **data limitations**: our existing in-house samples for groups larger than four often contain identities with highly similar poses, expressions, and low-quality faces. Incorporating such data leads the model to "copy-paste" humans, degrading the quality of the generated images. Including >4P scenarios with rigorous data filtering would improve performance. Second, **base model limitations**: FLUX Kontext itself is much less robust for >4P generation. Access to the raw FLUX.1 Kontext model, prior to high-quality finetuning or even before guidance distillation, would likely allow for room in the improved performance.

| Layered Canvas | Ours | Layered Canvas | Ours |

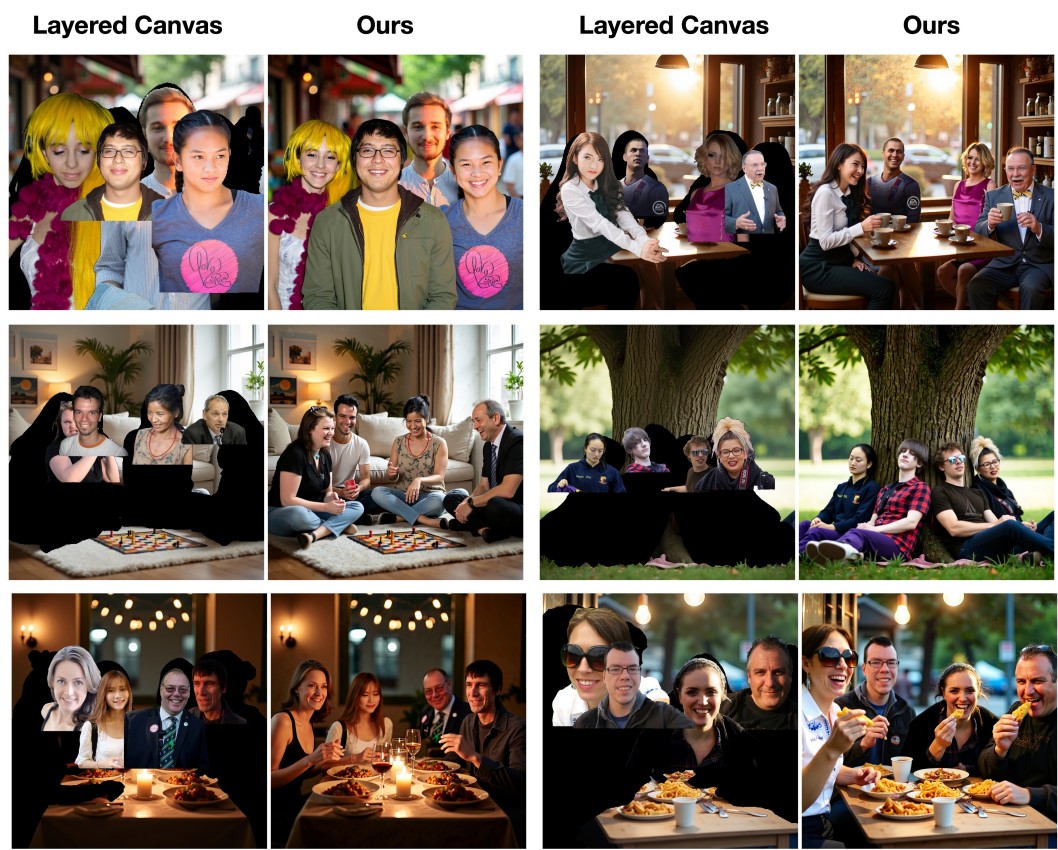

Figure I: **4P Personalization with Background.** Our layered canvas can be seamlessly integrated with the background, resulting in five layers: four persons and one background. In the final output, the inserted humans interact naturally with the background, *e.g.* leaning against a tree trunk or taking food from the table, while maintaining overall coherent lighting.

## C LLM USAGE DECLARATION

We only used large language models (LLMs) to polish the writing, *e.g.* correcting grammar and improving formality.

## D ADDITIONAL RESULTS

### D.1 ADDITIONAL 4P PERSONALIZATION WITH BACKGROUND

The layered canvas also accepts an optional background image as input. LayerComposer is able to generate images where humans interact naturally with the background under coherent lighting, as indicated in Fig. I.

### D.2 ADDITIONAL ABLATION STUDY

We provide another example illustrating the usefulness of locking, *e.g.* preserving the symbolic pose of an identity, as shown in Fig. II.

### D.3 ADDITIONAL 4P RESULTS

This section presents an extensive set of 32 qualitative examples for 4-person personalization, organized across four figures, to provide a comprehensive and uncurated evaluation of our method.

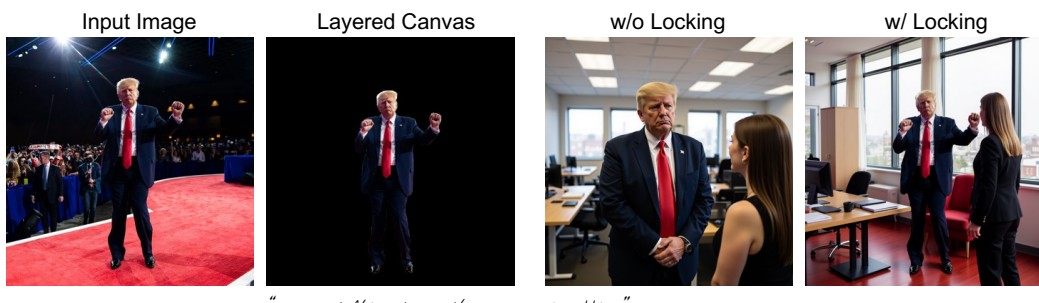

| Input Image | Layered Canvas | w/o Locking | w/ Locking |

*"a man talking to another woman in office"*

Figure II: **Additional Locking Example.** Locking can be used to preserve a subject's symbolic pose during generation.

We provide qualitative results in Tab. I. We compare our results against several key baselines: Overlay Kontext (ilkerzgi & gokaygokay, 2025), Qwen Image Edit (Wu et al., 2025a), and Nano-Banana (Comanici et al., 2025), with FLUX Kontext (Labs et al., 2025) serving as our base model.

Across these diverse scenarios, a clear pattern of performance emerges. Overlay Kontext often results in a simple 'cut-and-paste' look with poor semantic integration. Qwen Image Edit frequently alters the global scene context or fails to preserve subject identity. Nano-Banana (Comanici et al., 2025) is a noteworthy baseline; in the rare instances where it does not fail, it can produce highly naturalistic images that are free of artifacts. We believe this is a result of its stronger closed-model architecture and extensive data filtering. However, this peak performance is undermined by a critical lack of robustness. The model's performance is erratic, frequently failing entirely or producing images with significant compositional and anatomical artifacts. In stark contrast, our approach consistently generates high-quality, coherent images that faithfully adhere to input conditions, demonstrating a qualitative superiority rooted in robustness and reliability.

### D.4 ADDITIONAL 2P RESULTS

To demonstrate the effectiveness of our method on paired subjects, we provide a comprehensive set of qualitative examples for 2-person personalization in Tab. II. We compare our approach against several state-of-the-art methods following our experiments provided in the main paper: UniPortrait (He et al., 2025), StoryMaker (Zhou et al., 2024), UNO (Wu et al., 2025c), and OmniGen2 (Wu et al., 2025b).

The results highlight different failure modes among the baselines. UniPortrait (He et al., 2025) often struggles with identity preservation and produces an unnatural, 'pasted-on' effect. StoryMaker (Zhou et al., 2024) frequently generates overly stylized or semantically incoherent images, leading to failure cases. Although UNO (Wu et al., 2025c) and OmniGen2 (Wu et al., 2025b) show stronger performance, they can suffer from inconsistent subject interaction and subtle identity drift. In contrast, our method consistently excels at preserving subject identities, rendering realistic interactions, and maintaining coherence across a wide variety of challenging scenarios, underscoring its superior qualitative performance and reliability.

### D.5 ADDITIONAL 1P RESULTS

To showcase the fidelity of our method for single person personalization, this section provides a comprehensive set of qualitative comparisons in Tab. III. We benchmark our approach against several powerful, recent methods: IP-Adapter(IPA-FLUX) (Ye et al., 2023), PuLID (Guo et al., 2024), and InfiniteYou (Jiang et al., 2025).

The results reveal critical distinctions in model capability and reliability. IPA-FLUX (Ye et al., 2023) often suffers from failure cases or severe identity leakage, failing to disentangle the input identity from the subject in the original scene. Although PuLID-FLUX (Guo et al., 2024) and InfiniteYou (Jiang et al., 2025) are strong baselines that preserve identity more reliably, they can lack fidelity in other areas. PuLID-FLUX occasionally struggles with producing natural poses and expressions, while InfiniteYou can produce results with a level of diversity that falls short in reflecting the input condition.

In contrast, our method consistently achieves a higher degree of realism, excelling at preserving the subject's identity in a flexible way while seamlessly integrating them into the scene with appropriate lighting, texture, and pose.

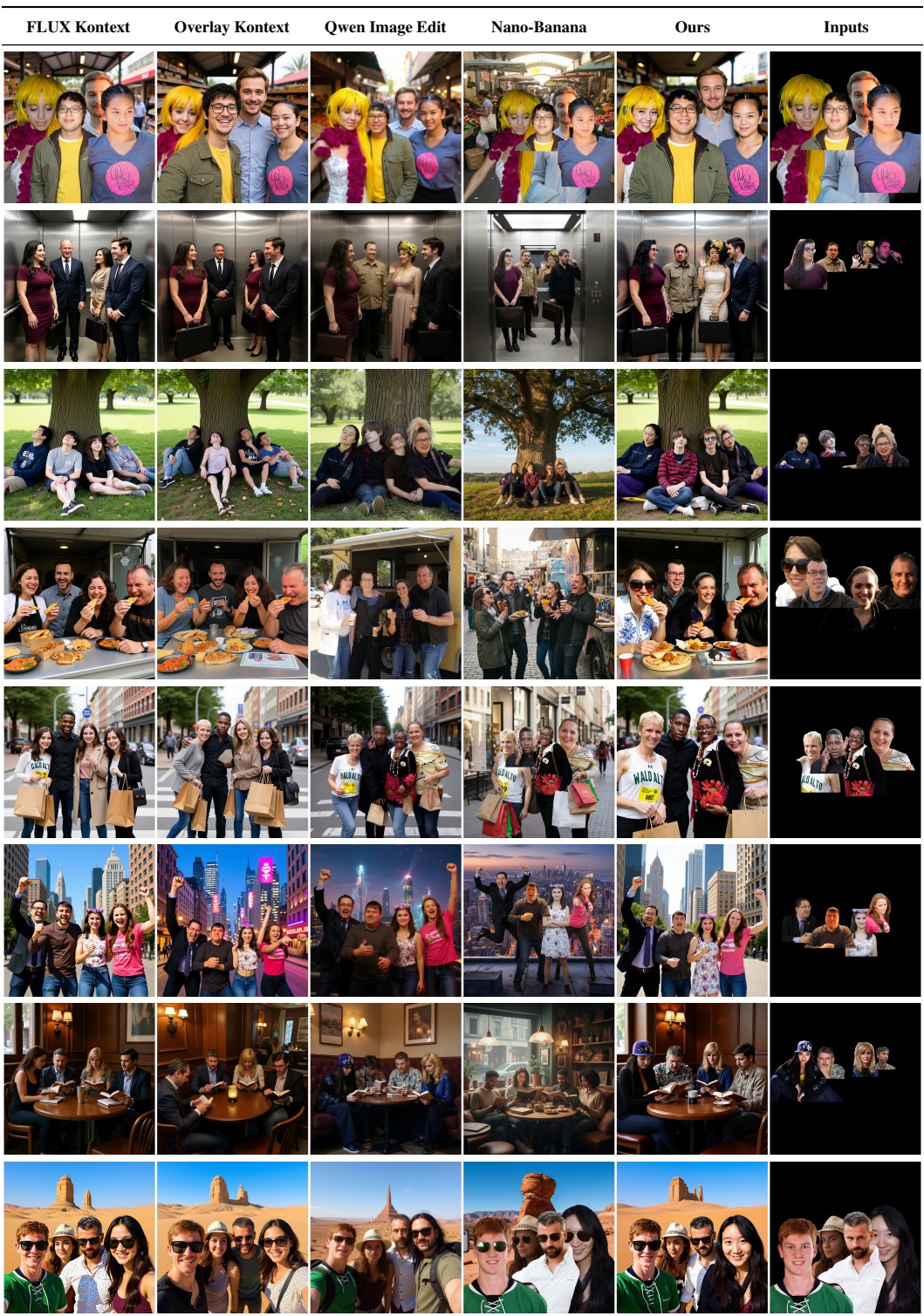

Table I: Supplementary results for 4P personalization

| UniPortrait | StoryMaker | UNO | OmniGen2 | Ours | Inputs |
| --- | --- | --- | --- | --- | --- |

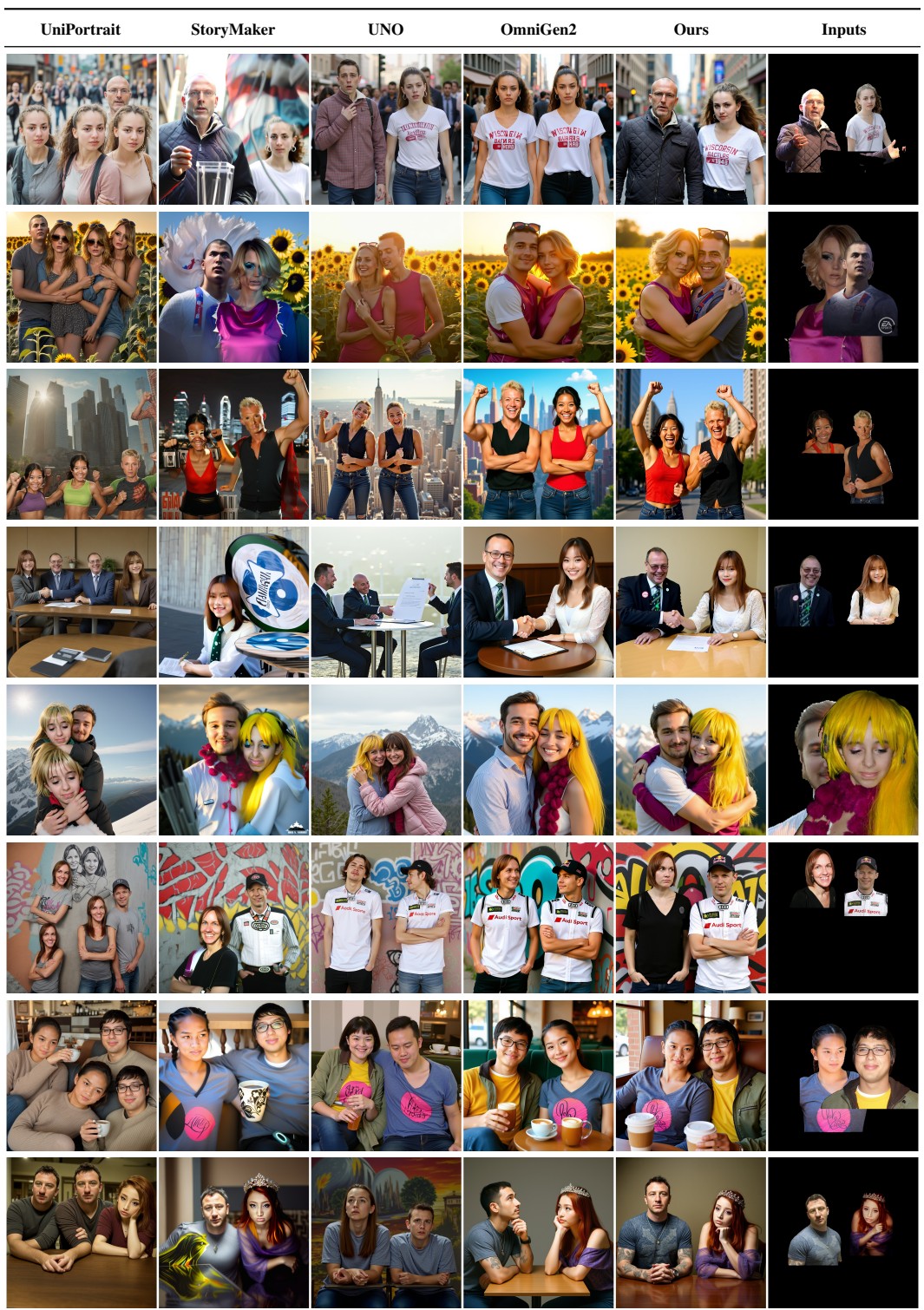

Table II: Supplementary results for 2P personalization

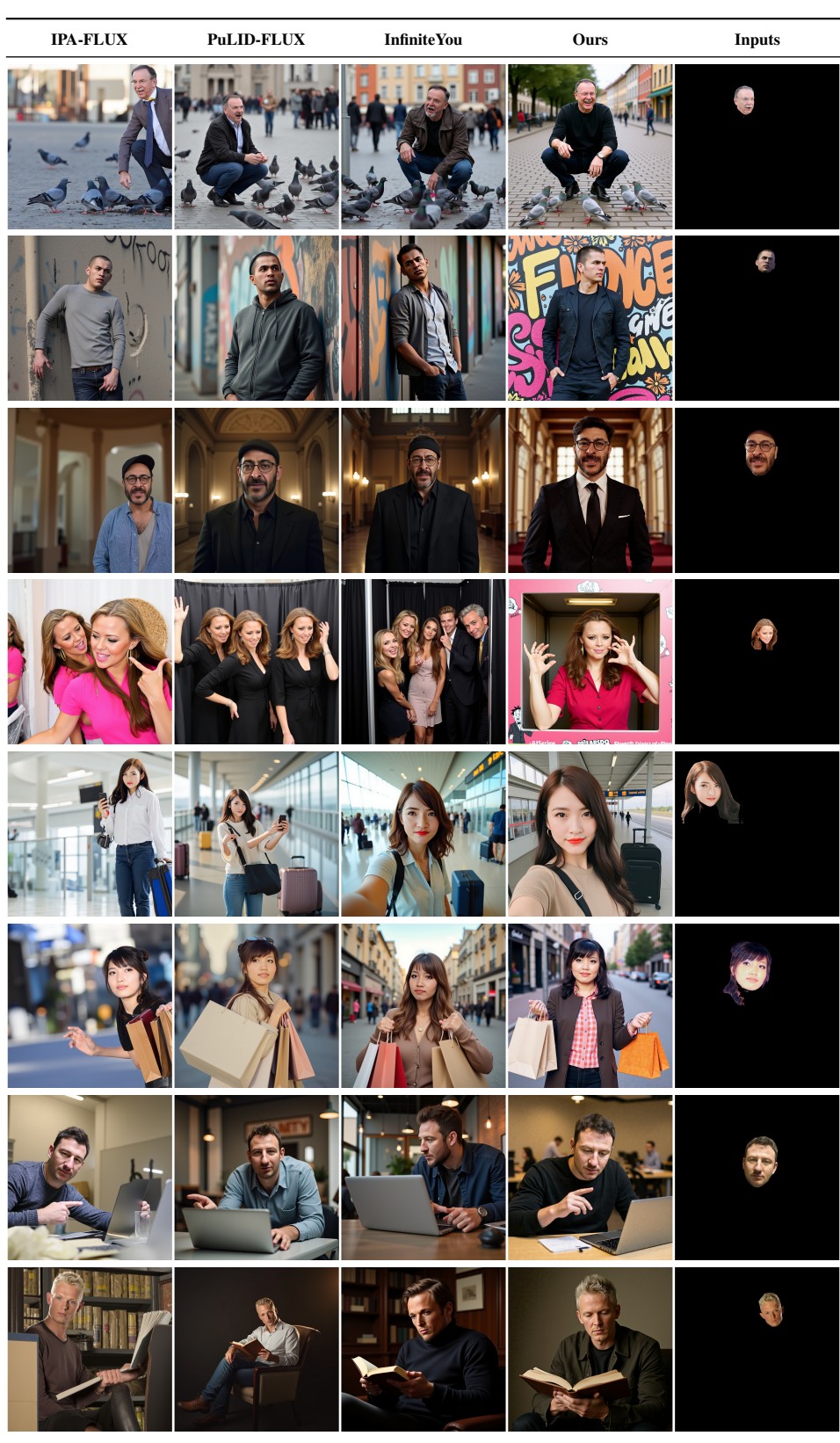

Table III: Supplementary results for 1P personalization

