# OpenReview forum: "LayerComposer: Interactive Personalized T2I via Spatially-Aware Layered Canvas"
_ICLR.cc/2026/Conference — ICLR 2026 Conference Withdrawn Submission_

### Official Review · Reviewer_jdiU · 2025-10-28

**Soundness:** 3
**Presentation:** 3
**Contribution:** 2
**Rating:** 4
**Confidence:** 4

**Summary:**

This paper introduces LayerComposer, a novel framework for interactive, multi-subject, personalized text-to-image generation. The core contributions are twofold: (1) a "layered canvas" representation that allows users to place subjects on distinct, spatially-aware layers, akin to professional editing software, which naturally handles occlusions; and (2) a "locking mechanism" that enables high-fidelity preservation of selected layers (e.g., a specific person's pose or the background) while allowing unlocked layers to be synthetically adapted to the scene and text prompt. The method is built upon a pretrained DiT model and employs a "transparent latent pruning" strategy for computational efficiency. The authors conduct extensive experiments on 1-person, 2-person, and 4-person personalization benchmarks, demonstrating qualitatively and quantitatively performance against several state-of-the-art methods.

**Strengths:**

1. The paper addresses a well-recognized problem in personalized image generation—the difficulty of fine-grained spatial control. The proposed layered canvas paradigm is a practical and user-centric approach.
2. The framework demonstrates an ability to generate coherent compositions that often maintain subject identity and spatial arrangement more reliably than some of the compared baselines.
3. The locking mechanism allows for a degree of content preservation without requiring architectural modifications to the base model.

**Weaknesses:**

1. The baseline models were almost certainly trained on publicly available, general-purpose datasets (e.g., LAION). Comparing a model fine-tuned on 32M private, domain-specific (human-centric) images against these baselines is an inequitable evaluation. The performance gap reported is likely inflated due to this data advantage instead of the method's contribution.
2. The quantitative analysis in Table 1 is insufficient and raises concerns. The more common and arguably more important 1P and 2P benchmarks are where the evaluation falls short. For these tasks, the comparison set seems carefully curated to exclude the most relevant and powerful class of competing models.
3. This lack of a proper comparison casts serious doubt on the practical utility of the proposed framework for common use cases.

**Questions:**

See weakness.

---

### Official Review · Reviewer_PtdL · 2025-10-28

**Soundness:** 3
**Presentation:** 3
**Contribution:** 3
**Rating:** 4
**Confidence:** 4

**Summary:**

This paper introduces a novel approach for multi-subject composition tasks, specifically designed to handle occlusions between subjects. The authors conceptualize the input as a series of layered canvases. A "locking embedding" is introduced for each layer, allowing the model to selectively prioritize which layers should be treated as high-fidelity conditions. The proposed method demonstrates superior qualitative and quantitative results, surpassing baseline methods on multi-subject composition benchmarks.

**Strengths:**

1. **High-Quality Composition:** The method produces impressive qualitative results in multi-character scenarios, effectively managing complex interactions compared to baseline methods.

2. **Targeted Design for Occlusion:** The paper proposes a specific and novel layered-canvas design to address the challenging problem of composing multiple, potentially occluding, characters.

**Weaknesses:**

1. **Insufficient Ablation Studies:** The paper lacks detailed ablation studies on key design components of the pipeline, such as the transparent latent pruning mechanism. This makes it difficult to assess the individual contribution of each component.

2. **Limited Validation of the Locking Mechanism:** The layer locking mechanism, highlighted as a core contribution, is not sufficiently validated through rigorous experiments and ablation studies.

**Questions:**

1. **Inconsistent Occlusion in Figure 7:** In Figure 7, the occlusion relationship between character 1 and character 2 appears to change across the "lock 1," "lock 1 & 2," and "lock 1 & 2 & 3" examples. Could you please explain the underlying reason for this inconsistency in the occlusion hierarchy?

2. **Need for Quantitative Posture and Position Evaluation:** A noticeable drift in the position and posture of characters occurs, even when the locking mechanism is applied. This suggests that fidelity is not perfectly maintained.

    - I strongly recommend that the authors quantitatively evaluate the posture and position consistency of the generated results. Metrics such as mean Average Precision (mAP) on pose landmarks for posture alignment and mean Intersection over Union (mIoU) of bounding boxes for positional alignment would be suitable.

    - Could you provide an analysis of why this phenomenon occurs, linking it back to the model's design and training strategy?

3. **Clarification on Training Strategy:** In Section 3.3, author(s) state that "unlocked layers provide cross-image appearance references without pixel-level correspondence." Could you please elaborate on this statement? Specifically, how are unlocked instances selected and utilized during the training process?

4. **Insufficient Ablation of the Layer Locking Mechanism:** The empirical evidence supporting the effectiveness of the locking mechanism is sparse. There are no quantitative results presented for its impact, and its qualitative benefits are demonstrated in only two figures (Figure II and Figure 7). To properly validate this core contribution, a more thorough ablation study is necessary. Could you please provide an analysis, including both quantitative metrics (e.g., posture/position alignment) and visual examples, for the following scenarios?

    - Multiple characters with only one character locked.

    - Multiple characters with a subset of characters locked (with and without significant occlusion).

    - Multiple characters, all locked (with and without significant occlusion).

5. **Generalization to Non-Human Subjects:** The paper's examples predominantly feature human subjects. Only Figure 1 includes a non-human character (a 3D cartoon). Could you provide more examples to demonstrate the method's effectiveness and generalization capabilities on a wider range of non-human subjects (e.g., animals, objects, different art styles)?

---

### Official Review · Reviewer_HBCa · 2025-10-31

**Soundness:** 3
**Presentation:** 3
**Contribution:** 2
**Rating:** 4
**Confidence:** 5

**Summary:**

LayerComposer presents an interactive framework for personalized text-to-image generation that enables fine-grained control over multi-subject composition. It introduces a layered canvas where each subject is placed on a separate RGBA layer, allowing occlusion-free arrangement and flexible editing. A locking mechanism helps users preserve specific subjects (e.g., background or individuals) with high fidelity—retaining identity while allowing natural lighting and context integration—while other unlocked subjects are synthesized according to the text prompt. This is achieved through a model-data co-design using positional embeddings and a locking-aware data sampling strategy, without modifying the underlying model architecture. Experiments show that LayerComposer outperforms state-of-the-art methods in spatial control, identity preservation, and compositional flexibility, especially in complex multi-person scenarios.

**Strengths:**

1. The overall writing is easy to follow.
2. The layered canvas paradigm offers a Photoshop-like interface for multi-subject composition, enabling intuitive control over placement, scaling, and locking of subjects—significantly improving creative interactivity.
3. The overall framework design is intuitive and well-justified, enabling the model to simultaneously learn to preserve faithful details from reference inputs (hard copy) and flexibly compose scenes according to the text prompt.
4. The authors provide extensive visual results on multi-character composition to demonstrate the robustness and effectiveness of their method across diverse scenarios, including complex compositions and occluded inputs.

**Weaknesses:**

1. Limited Novelty and Questionable Effectiveness: The task setup closely follows prior work such as CollageDiffusion [1] and NoiseCollage [2], both of which have already introduced the concept of layered composition for multi-concept generation. LayerComposer builds upon Flux Kontext—a strong in-context editing model with inherent spatial and identity priors—through fine-tuning, making it difficult to disentangle whether the improved performance stems from the proposed framework or simply from leveraging a powerful pretrained backbone. As a result, the claimed novelty and effectiveness of the layered canvas and locking mechanism are not sufficiently substantiated.
2. Unfair Comparison with SOTA Editing Methods: The evaluation compares LayerComposer against strong in-context editing models such as Kontext, Qwen-Image-Edit, and Nano-Banana, which are designed to edit or extend input images based on natural language instructions. These methods operate under minimal user guidance and infer spatial layout, semantic coherence, and identity preservation from limited input. In contrast, LayerComposer takes a multi-layer canvas with explicit spatial arrangements, masks, and locking flags—providing far richer structural priors. Its superior performance in composition and alignment may therefore stem largely from this privileged, hand-crafted input rather than intrinsic modeling advances. This fundamental asymmetry in input control makes the comparison misleading and overstates the method’s advantage.
3. Limited Generalization Beyond Human Subjects: The paper only evaluates the method on human-centric composition tasks, showcasing results primarily involving people in social scenes. This narrow focus makes it difficult to assess the generalizability of LayerComposer to more diverse object categories—such as animals, furniture, or vehicles—where challenges like complex occlusions, viewpoint variations, and geometric perspective become significantly more pronounced. The framework’s reliance on segmentation masks and spatial alignment may not extend robustly to non-rigid or highly articulated objects, yet the authors do not discuss these limitations or provide any experiments beyond human subjects. As a result, the method’s broader applicability remains unproven.
4. Lighting Inconsistency Caused by the Locking Mechanism: The locking mechanism preserves most visual details of input images (e.g., texture, shading, and local lighting), which can lead to severe illumination mismatches when composing multiple subjects into a single scene—especially in multi-person settings. As shown in Figure 7, locked subjects retain their original lighting conditions, resulting in inconsistent shadows, color tones, and highlights across individuals in the final image. This undermines the global coherence of the scene and degrades generation quality, indicating that the model lacks effective mechanisms for appearance harmonization under strong input constraints.


[1] Sarukkai V, Li L, Ma A, et al. Collage diffusion[C]//Proceedings of the IEEE/CVF winter conference on applications of computer vision. 2024: 4208-4217.

[2] Shirakawa T, Uchida S. Noisecollage: A layout-aware text-to-image diffusion model based on noise cropping and merging[C]//Proceedings of the IEEE/CVF conference on computer vision and pattern recognition. 2024: 8921-8930.

**Questions:**

Please refer to the weakness.

---

### Official Review · Reviewer_AH3R · 2025-10-31

**Soundness:** 3
**Presentation:** 3
**Contribution:** 2
**Rating:** 4
**Confidence:** 5

**Summary:**

LayerComposer introduces an interactive, scalable, and spatially-aware framework for multi-subject personalized text-to-image generation via a layered canvas and locking mechanism.

**Strengths:**

- Novel problem: This work may be the first to address multi-identity personalization within a specified scene, providing a practical direction for controllable multi-subject generation.
- Scalability and efficiency: The transparent latent pruning strategy effectively decouples token sequence length from the number of subjects, overcoming scalability bottlenecks found in prior methods.
- Elegant engineering: The use of a 6M multi-image-per-scene dataset and a straightforward multi-layer design built upon FLUX-Kontext makes the approach both simple and practical in implementation.

**Weaknesses:**

- Missing related work: Highly relevant open-source works such as ID-Patch [1] and DreamO [2], which also focus on multi-identity personalization, are neither cited nor compared.
- Limited comparison: In the 4-person (4P) setting, the evaluation includes only image-editing models, while other multi-ID personalization methods (e.g., UniPortrait, ID-Patch, DreamO) that support similar settings are omitted. Moreover, using a canvas input may disadvantage editing models, as LayerComposer is optimized for this input format whereas editing models are not designed for all-black backgrounds.
- Limited novelty and reproducibility: Despite the impressive dataset scale and engineering effort, the method essentially functions as a large LoRA fine-tuned on FLUX-Kontext with intuitive solutions. The use of an in-house 32M-image dataset that is not publicly available raises concerns about reproducibility and fairness.
References
[1] Zhang, Yimeng, et al. "ID-Patch: Robust ID association for group photo personalization." CVPR, 2025.
 [2] Mou, Chong, et al. "DreamO: A unified framework for image customization." arXiv preprint arXiv:2504.16915, 2025.

**Questions:**

- The evaluation employs different model sets across 1P, 2P, and 4P tasks. Why editing models in 4P setting are not tested in 2P or 1P settings? And why methods in 2P settings are not tested in 4P settings? The authors argue that existing methods "struggle to scale," yet the absence of such comparisons makes this claim insufficiently supported.

---

### Note · Authors · 2025-11-12

I have read and agree with the venue's withdrawal policy on behalf of myself and my co-authors.